# Expressive Pooling for Graph Neural Networks

**Veronica Lachi**[*]                                                                    *vlachi@fbk.eu*
*Fondazione Bruno Kessler*

**Alice Moallemy-Oureh**[*]                                                      *amoallemy@uni-kassel.de*
*University of Kassel*

**Andreas Roth**[*]                                                          *andreas.roth@tu-dortmund.de*
*TU Dortmund*

**Pascal Welke**[*]                                                              *p.welke@lancaster.ac.uk*
*Lancaster University Leipzig and TU Wien*

**Reviewed on OpenReview:** *https://openreview.net/forum?id=xGADInGWMt*

## Abstract

Considerable efforts have been dedicated to exploring methods that enhance the expressiveness of graph neural networks. Current endeavors primarily focus on modifying the message-passing process to overcome limitations imposed by the Weisfeiler-Leman test, often at the expense of increasing computational cost. In practical applications, message-passing layers are interleaved with pooling layers for graph-level tasks, enabling the learning of increasingly abstract and coarser representations of input graphs. In this work, we formally prove two directions that allow pooling methods to increase the expressive power of a graph neural network while keeping the message-passing method unchanged. We systematically assign eight frequently used pooling operators to our theoretical conditions for increasing expressivity. Experiments conducted on the Brec dataset confirm that those pooling methods that satisfy our conditions empirically increase the expressivity of graph neural networks.

## 1 Introduction

With the growing availability of graph data and the success of neural networks, graph neural networks (GNNs) (Scarselli et al., 2008) have emerged as a key research area. Most common among GNNs, message-passing neural networks (MPNNs) update node representations by aggregating information from neighbors, analogous to the 1-Weisfeiler-Leman (1-WL) coloring test used for graph isomorphism. This similarity limits MPNNs' expressive power to that of the 1-WL test (Maron et al., 2019; Morris et al., 2019; Xu et al., 2019). MPNNs like the graph isomorphism network (GIN) (Xu et al., 2019) improve aggregation functions to achieve the same expressivity as 1-WL. This limitation has prompted the design of GNNs that do not follow this message-passing scheme (Zhang et al., 2023). Advancements include k-GNNs (Morris et al., 2019), which use higher-order message passing, and methods like ESAN (Bevilacqua et al., 2022) and GSN (Bouritsas et al., 2023) that encode subgraph structures. Zhang et al. (2023, Table 2) provide an overview including these approaches. These are often referred to as higher-order GNNs.

Jogl et al. (2023) show that MPNNs can achieve the same expressivity as higher-order GNNs by simulating higher-order GNNs with MPNNs using graph transformations. They prove that many recent higher-order GNN models, including those by Morris et al. (2019; 2020; 2022); Qian et al. (2022), can be simulated by MPNNs on transformed graphs. However, all graph transformations presented by Jogl et al. (2023) increase the input graph size and consequently incur a higher computational cost for the resulting subsequent message-

---

[*]All authors contributed equally.

passing phase. Similarly, the computational cost of non-simulated higher-order GNNs increases in practice and in asymptotic runtime analysis, compared to MPNNs.

Pooling operators (Ying et al., 2018; Diehl, 2019; Gao & Ji, 2019; Lee et al., 2019; Ranjan et al., 2020), on the other hand, aim to coarsen graphs by reducing the number of nodes or edges. Here, MPNNs are interleaved with graph transformations which *decrease* the graph size. By combining sets of nodes into a single supernode, MPNNs can identify relations between coarsened structures and combine the features of distant nodes. These also aim to improve the runtime of learning or inference and have achieved practical predictive performance improvements (Bianchi et al., 2020b; Ying et al., 2018; Liu et al., 2023). This line of work has long remained disconnected from expressivity research. Only recently, Bianchi & Lachi (2023) established conditions under which pooling operators retain the expressivity of an MPNN. It remains unclear if and under which conditions adding a pooling operator can increase the expressivity of an MPNN.

To our knowledge, no pooling operator that decreases graph size has yet been designed to *increase* expressivity. Conversely, we are not aware of any expressivity-increasing GNN that reduces the graph size when considering its underlying graph transformation. It seems natural that improving the expressive power of message passing requires increased computational resources and that, hence, the goals of pooling operators and higher-order GNNs are incompatible. In this work, we refute this assumption. We identify two sufficient conditions for pooling operators to increase the expressivity of MPNNs. First, expressivity can be improved by utilizing a sufficiently expressive node selection operation that identifies structurally different sets of nodes. Second, we prove that local pooling operators can also construct WL distinguishable coarsened graphs by only considering node features and their connectivity. As a negative result, we prove that pooling operators that only utilize node features obtained by MPNNs cannot improve expressivity. We survey existing pooling operators that satisfy the positive or negative conditions and present multiple illustrative pooling operators. Furthermore, we construct an exemplary, practical pooling operator XP that increases the expressivity of MPNNs and experimentally confirm on synthetic datasets that XP enhances the expressivity of MPNNs. Our positive conditions and their practical implementation in XP may serve as guidelines for designing further expressivity-increasing pooling methods in the future.

## 2 Preliminaries

Most of our notation is taken from Bianchi & Lachi (2023). We use $\{\!\{\cdot\}\!\}$ to denote multisets, i.e., unordered collections that allow repeated entries. Let $\mathcal{G} = (\mathcal{V}, \mathcal{E}, \mathbf{H})$ be a graph with $\mathcal{V}$ the set of nodes such that $n = |\mathcal{V}|$, $\mathcal{E}$ the set of edges, and $\mathbf{H} \in \mathbb{R}^{n \times d}$ a matrix of node features. For a given node $v$, the $v$-th row of $\mathbf{H}$ is denoted by $h_v$, and the set of neighboring nodes is denoted by $\mathcal{N}_v$.

A message-passing graph neural network (MPNN) is a neural network architecture computing a function in the graph domain. MPNNs provide a flexible framework and are end-to-end differentiable, allowing their parameters to be adapted to a given task using gradient descent. Formally:

**Definition 2.1** (MPNN). An MPNN is a neural network architecture that, at each iteration updates the embeddings of all nodes by aggregating their embeddings and those of their neighbors in the previous iteration. After $T \geq 0$ iterations, the node embeddings are aggregated into the graph embedding $x_G$:

$$x_v^{(t)} = \text{UPD}^{(t)}\left(x_v^{(t-1)}, \text{AGG}^{(t)}\left(\{\!\{x_u^{(t-1)} \; : \; u \in \mathcal{N}_v\}\!\}\right)\right) \qquad x_G = \text{READ}\left(\{\!\{x_v^{(T)} \; : \; v \in V\}\!\}\right) \quad (1)$$

with $x_v^{(0)} = h_v$ and $t \leq T$. $x_v^{(t)} \in \mathbb{R}^{d(t)}$ and $x_G \in \mathbb{R}^{d'}$ are the embedding of node $v$ after the $t$-th layer and the graph embedding, respectively.

MPNNs typically implement $\text{AGG}^{(t)}$ as a permutation invariant aggregation function and $\text{UPD}^{(t)}$ using a multilayer perceptron (MLP). Sum and mean pooling are popular for READ. When a method utilizes additional or other operations, we refer to those more generally as GNNs.

### 2.1 Expressivity of GNNs

When studying the expressive power of GNNs, the core aim is to understand their ability to distinguish non-isomorphic graphs. MPNNs are known to be limited by the Weisfeiler-Leman isomorphism test (Leman & Weisfeiler, 1968), i.e., they cannot distinguish graphs that this test cannot distinguish (Xu et al., 2019).

**Definition 2.2** (Weisfeiler-Leman test). The Weisfeiler-Leman test is an iterative node feature (color) refinement algorithm to test whether two graphs are isomorphic. Let $\Sigma$ be a set of node colors. At iteration 0, let $c_v^{(0)} = \text{HASH}_0(h_v)$, where HASH is an injective function that maps node features to colors in $\Sigma$. For any iteration $t$, let

$$c_v^{(t)} = \text{HASH}\left(c_v^{(t-1)}, \{\!\{c_u^{(t-1)} \; : \; u \in \mathcal{N}_v\}\!\}\right) \qquad\qquad c_G^{(t)} = \{\!\{c_v^{(t)} \; : \; v \in \mathcal{V}\}\!\}.$$

The algorithm terminates if the number of colors between two iterations does not change, or equivalently when there exists a bijection between $c_G^{(t-1)}$ and $c_G^{(t)}$.

**Definition 2.3** (WL distinguishable). $\mathcal{G}_1$ and $\mathcal{G}_2$ are **WL distinguishable** ($\mathcal{G}_1 \neq_{\text{WL}} \mathcal{G}_2$) if there exists an iteration $t$ for which $c_{\mathcal{G}_1}^{(t)} \neq c_{\mathcal{G}_2}^{(t)}$.

If two graphs are WL distinguishable, the WL algorithm (correctly) concludes that they are not isomorphic after termination. MPNNs are at most as expressive as the WL test in distinguishing between graphs (Xu et al., 2019) in the following sense:

**Definition 2.4.** Let $\varphi, \psi$ be permutation invariant functions from the set of graphs to a vector domain and $\mathcal{G}_1, \mathcal{G}_2, \mathcal{G}_3, \mathcal{G}_4$ be graphs. Then

- $\varphi$ is **at least as expressive as** $\psi$ if $\psi(\mathcal{G}_1) \neq \psi(\mathcal{G}_2) \Rightarrow \varphi(\mathcal{G}_1) \neq \varphi(\mathcal{G}_2)$.

- $\varphi$ and $\psi$ are **equally expressive** if $\psi(\mathcal{G}_1) \neq \psi(\mathcal{G}_2) \Leftrightarrow \varphi(\mathcal{G}_1) \neq \varphi(\mathcal{G}_2)$.

- $\varphi$ is **more expressive** than $\psi$, if $\varphi$ is at least as expressive as $\psi$ and there exist $\mathcal{G}_1, \mathcal{G}_2$ with $\varphi(\mathcal{G}_1) \neq \varphi(\mathcal{G}_2)$ and $\psi(\mathcal{G}_1) = \psi(\mathcal{G}_2)$.

- Finally, $\varphi, \psi$ are **incomparable** if there exist $\mathcal{G}_1, \mathcal{G}_2, \mathcal{G}_3, \mathcal{G}_4$ such that $\varphi(\mathcal{G}_1) \neq \varphi(\mathcal{G}_2)$ with $\psi(\mathcal{G}_1) = \psi(\mathcal{G}_2)$ and $\varphi(\mathcal{G}_3) = \varphi(\mathcal{G}_4)$ with $\psi(\mathcal{G}_3) \neq \psi(\mathcal{G}_4)$.

For example, the WL test is at least as expressive as any MPNN as defined in Equation (1) (Xu et al., 2019). To obtain an MPNN that is as expressive as the WL test, UPD and AGG need to be injective functions on the input domain. A suitable multilayer perceptron and sum aggregation achieve this (Xu et al., 2019).

In our subsequent discussion, we will analyze the application of graph pooling operators after $t$ iterations of message passing. To simplify notation, we will from now on ignore the number of such iterations. We will formulate our expressivity results for graphs $\mathcal{G} = (\mathcal{V}, \mathcal{E}, \mathbf{H})$ where $\mathbf{H} = \mathbf{X}^{(t)}$, i.e., the node features are obtained by running $t$ iterations of an MPNN. To compare the expressivity of such *composite* GNNs before and after applying a pooling operator, we introduce current distinguishability:

**Definition 2.5** (currently distinguishable). Two graphs $\mathcal{G}_1 = (\mathcal{V}_1, \mathcal{E}_1, \mathbf{H}_1)$ and $\mathcal{G}_2 = (\mathcal{V}_2, \mathcal{E}_2, \mathbf{H}_2)$ with (hidden) node representations $\mathbf{H}_1$ and $\mathbf{H}_2$ are **currently distinguishable** (with respect to $\mathbf{H}_1$ and $\mathbf{H}_2$), denoted as $\mathcal{G}_1 \neq_{CD} \mathcal{G}_2$, if their multisets of node representations $\{\!\{\mathbf{h}_{1_v} \; : \; v \in \mathcal{V}_1\}\!\} \neq \{\!\{\mathbf{h}_{2_v} \; : \; v \in \mathcal{V}_2\}\!\}$ are different.

Note that currently distinguishable graphs, by definition, are WL distinguishable.

### 2.2 Pooling in GNNs

Despite the large body of work on the expressive power of GNNs, insights into the interplay between GNNs and pooling operators are missing. In this work, we analyze the capabilities of pooling operators to enhance the expressive power of GNNs. We follow Grattarola et al. (2022) and write pooling operators as a triplet (`SEL, RED, CON`) of Select-Reduce-Connect functions.

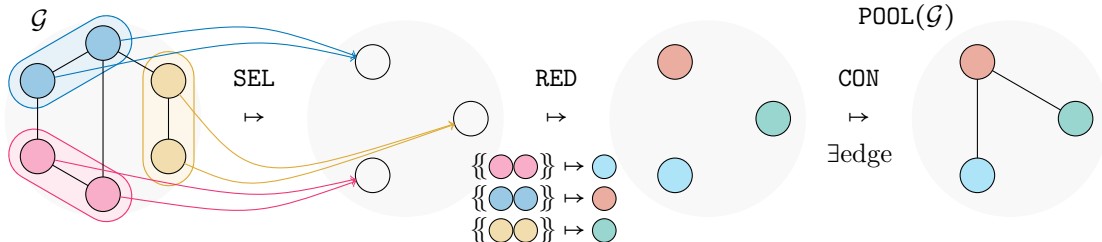

Figure 1: Example of a pooling operator that clusters all nodes of identical color and checks if there exit edges between the clusters: SEL creates a supernode for each node color appearing in $\mathcal{G}$. RED assigns new colors to the supernodes, hashing each individual color multiset to a new color. CON checks if there exit edges between nodes of different colors in $\mathcal{G}$ and sets an edge between supernodes in POOL($\mathcal{G}$).

**Definition 2.6** (SEL, RED, CON functions). A pooling operator POOL $: \mathcal{G} \mapsto \mathcal{G}^P := (\mathcal{V}^P, \mathcal{E}^P, \mathbf{H}^P)$ is a function that maps a graph to a new graph with $|\mathcal{V}^P| \leq |\mathcal{V}|$. POOL can be written as select-reduce-connect triplet.

- SEL  The select function SEL $: \mathcal{G} \mapsto \mathbf{S} \in \mathbb{R}^{n \times k}$ clusters the input graph nodes into $k \geq 1$ so-called supernodes $\mathbf{S}_j = (\mathbf{s}_i^j)_{i=1}^n$ where $\mathbf{s}_i$ is the vector containing the contributions of node $i$ to all the supernodes and $\mathbf{s}_i^j$ indicates the contribution of node $i$ on supernode $j$.

- RED  The reduce function RED $: (\mathbf{S}, \mathbf{H}) \mapsto \mathbf{H}^P$ aggregates the features of the nodes assigned to the same supernode to produce its features.

- CON  The connect function CON $: (\mathcal{G}, \mathbf{S}) \mapsto \mathcal{E}^P$ generates the edges, by connecting the supernodes $\mathbf{S}$.

See Figure 1 for an example.

While various pooling operators have been proposed (Ying et al., 2018; Luzhnica et al., 2019; Bianchi et al., 2020a;b; Fey et al., 2020; Sanders et al., 2023; Tsitsulin et al., 2023), their impact on the expressive power of GNNs with respect to the WL test has largely remained unexplored. To the best of our knowledge, no prior work has analyzed conditions on the SEL, RED, and CON functions that allow pooling to *increase* expressivity beyond that of the underlying MPNN. The recent work by Bianchi & Lachi (2023) introduced a first formal treatment of expressivity in the context of pooling, establishing *sufficient conditions* under which a pooling operator *preserves* the expressive power of a given GNN. In this paper, we go one step further: we demonstrate that, under appropriate assumptions, pooling can in fact *enhance* the expressive power of the network, enabling it to distinguish graphs that the original MPNN could not.

## 3 Pooling Maintains Expressivity

We start by providing our formal definition of pooling operators that maintain expressivity.

**Definition 3.1** (Maintaining Expressivity). A pooling operator POOL = (SEL, RED, CON) is **maintaining expressivity** if it maps any pair of currently distinguishable graphs to a pair of currently distinguishable graphs, i.e., if $\mathcal{G}_1 \neq_{CD} \mathcal{G}_2 \Rightarrow$ POOL($\mathcal{G}_1$) $\neq_{CD}$ POOL($\mathcal{G}_2$).

When using WL or MPNN iterations, it is important to highlight that maintaining expressivity is meaningful when WL distinguishable graphs $\mathcal{G}_1$ and $\mathcal{G}_2$ have different node representations $\mathbf{H}_1$ and $\mathbf{H}_2$ or colors assigned. Indeed, if an insufficient number of WL or MPNN iterations has been performed, the graphs may be WL distinguishable in principle, yet their node representations are equal. In this case, regardless of the power of the pooling operator, the resulting graphs may remain indistinguishable.

Bianchi & Lachi (2023) established three conditions on the SEL and RED functions, which are sufficient to ensure that a GNN incorporating a pooling layer maintains the expressive power of the GNN without pooling. As a first contribution, we generalize these properties. In what follows, we show that the discriminative power of the WL test after pooling is maintained when injectivity of the reduce step is ensured.

**Proposition 3.2.** *Let $\mathcal{G} = (\mathcal{V}, \mathcal{E}, \mathbf{H})$ be a graph. Let $j \in \mathcal{V}_P$ be a supernode of a pooled graph $\texttt{POOL}(\mathcal{G})$. Let*

$$\left[\texttt{RED}(\mathbf{S}, \mathbf{H})\right]_j \coloneqq f\left(\{\!\{(\mathbf{h}_v, s_v^j) \; : \; v \in \mathcal{V}\}\!\}\right)$$

*where $f$ is injective on multisets. Then, $\texttt{POOL} = (\texttt{RED}, \texttt{SEL}, \texttt{CON})$ is maintaining expressivity.*

*Proof.* Let $\mathcal{G}_1 = (\mathcal{V}_1, \mathcal{E}_1, \mathbf{H}_1)$ and $\mathcal{G}_2 = (\mathcal{V}_2, \mathcal{E}_2, \mathbf{H}_2)$ be two currently distinguishable graphs w.r.t. $\mathbf{H}_1$ and $\mathbf{H}_2$. Let $\texttt{POOL}(\mathcal{G}_1) = (\mathcal{V}_1^P, \mathcal{E}_1^P, \mathbf{H}_1^P)$ and $\texttt{POOL}(\mathcal{G}_2) = (\mathcal{V}_2^P, \mathcal{E}_2^P, \mathbf{H}_2^P)$ be the corresponding pooled graphs. We denote by $\mathcal{H}_1 \coloneqq \{\!\{\mathbf{h}_v \; : \; v \in \mathcal{V}_1\}\!\}$ the multiset of node representations given by $\mathbf{H}_1$. Analogously, $\mathcal{H}_2, \mathcal{H}_1^P, \mathcal{H}_2^P$. Now assume, for contradiction, that $\texttt{POOL}(\mathcal{G}_1)$ and $\texttt{POOL}(\mathcal{G}_2)$ are not currently distinguishable w.r.t. $\mathbf{H}_1^P$ and $\mathbf{H}_2^P$. That is, we have

$$\mathcal{H}_1^P = \{\!\{f\left(\{\!\{(\mathbf{h}_v, s_v^j) \; : \; v \in \mathcal{V}_1\}\!\}\right) \; : \; j \in \mathcal{V}_1^P\}\!\} = \{\!\{f\left(\{\!\{(\mathbf{h}_v, s_v^j) \; : \; v \in \mathcal{V}_2\}\!\}\right) \; : \; j \in \mathcal{V}_2^P\}\!\} = \mathcal{H}_2^P \; .$$

As $f$ is injective, it follows that

$$\{\!\{\{\!\{(\mathbf{h}_v, s_v^j) \; : \; v \in \mathcal{V}_1\}\!\} \; : \; j \in \mathcal{V}_1^P\}\!\} = \{\!\{\{\!\{(\mathbf{h}_v, s_v^j) \; : \; v \in \mathcal{V}_2\}\!\} \; : \; j \in \mathcal{V}_2^P\}\!\} \; . \tag{2}$$

By our initial assumption, we have $\mathcal{H}_1 \neq \mathcal{H}_2$. We make a case distinction:

**There exists an element $\mathbf{h} \in \mathcal{H}_1$ with $h \notin \mathcal{H}_2$.** Then it follows that the equality in Equation (2) cannot hold: For every supernode $j \in \mathcal{V}_1^P$, we have $(\mathbf{h}, \cdot) \in \{\!\{(\mathbf{h}_w, s_w^j) \; : \; w \in \mathcal{V}_1\}\!\}$, but no node $j' \in \mathcal{V}_2^P$ with $(\mathbf{h}, \cdot) \in \{\!\{(\mathbf{h}_w, s_w^{j'}) \; : \; w \in \mathcal{V}_2\}\!\}$. This implies that the equality in Equation (2) does not hold.

**There exists an element $\mathbf{h}$ that appears $k$ times in $\mathcal{H}_1$ and $l \neq k$ times in $\mathcal{H}_2$.** Then, we have

$$\left|\{\!\{\{\!\{(\mathbf{h}_w, s_w^j) \; : \; w \in \mathcal{V}_1\}\!\} \; : \; j \in \mathcal{V}_1^P \wedge \mathbf{h}_w = \mathbf{h}\}\!\}\right| = k \neq l = \left|\{\!\{\{\!\{(\mathbf{h}_w, s_w^j) \; : \; w \in \mathcal{V}_2\}\!\} \; : \; j \in \mathcal{V}_2^P \wedge \mathbf{h}_w = \mathbf{h}\}\!\}\right| \; .$$

which implies that the equality in Equation (2) does not hold.

Both cases result in a contradiction to the claimed equality of multisets $\mathcal{H}_1^P$ and $\mathcal{H}_2^P$, hence it follows that $\texttt{POOL}(\mathcal{G}_1) \neq_{\mathrm{CWL}} \texttt{POOL}(\mathcal{G}_2)$ must instead hold. $\qquad\square$

We note that this proof is not specifically formulated for node representations obtained by WL or MPNNs, and the expressivity is maintained even when the original representations are more discriminative. Based on Xu et al. (2019), injectivity on multisets can be achieved by combining the sum aggregation with an element-wise function:

**Proposition 3.3.** *Let $\mathcal{G} = \{\mathcal{G}_k \mid k \in \mathbb{N}\}$ be a countable set of graphs with $\mathcal{G}_k = (\mathcal{V}_k, \mathcal{E}_k, \mathbf{H}^{(k)})$ and a bounded number of nodes. For any $\texttt{SEL}(\mathcal{G}_k) = \mathbf{S}^{(k)}$, there exists a function $\phi: \mathbb{R}^d \times \mathbb{R} \to \mathbb{R}^{d'}$ such that*

$$\left[\texttt{RED}(\mathbf{S}^{(k)}, \mathbf{H}^{(k)})\right]_j \coloneqq \sum_{v \in \mathcal{V}_k} \phi((\mathbf{H}^{(k)})_v, (\mathbf{S}^{(k)})_v^j)$$

*is injective over all multisets $\{\!\{((\mathbf{H}^{(k)})_v, (\mathbf{S}^{(k)})_v^j) \; : \; v \in \mathcal{V}_k\}\!\}$.*

*Proof.* As the union of countably many sets of bounded size is countable, this proof is analogous to (Xu et al., 2019), which proves injectivity on multisets from a countable space. There exists a mapping $Z: ((\mathbf{H}^{(k)})_v, (\mathbf{S}^{(k)})_v^j) \mapsto \mathbb{N}$ that assigns a different natural number to each unique node element $((\mathbf{H}^{(k)})_v, (\mathbf{S}^{(k)})_v^j)$. Then, the function $\phi((\mathbf{H}^{(k)})_v, (\mathbf{S}^{(k)})_v^j) = n^{-Z((\mathbf{H}^{(k)})_v, (\mathbf{S}^{(k)})_v^j)}$, where $n$ is the maximum number of nodes in a graph, can be seen as a continuous one-hot encoding. Thus, any sum $\sum_{v \in \mathcal{V}_k} \phi((\mathbf{H}^{(k)})_v, (\mathbf{S}^{(k)})_v^j)$ is an injective function on multisets. $\qquad\square$

We will refer to this choice of $\texttt{RED}$ as M-$\texttt{RED}$ as it *maintains* expressivity. We can model $\phi$ by an MLP for all practical purposes due to the universal approximation theorem (Hornik et al., 1989). From now on, we assume that the $\texttt{RED}$ employed in the pooling operator adheres to the previously defined M-$\texttt{RED}$, ensuring the fulfillment of the maintaining expressivity condition. We note that this result, as in Xu et al. (2019),

holds in a non-uniform sense: the MLP used to approximate $\phi$ depends on the function being learned and may need to grow in size (i.e., number of parameters) with the size or complexity of the input graphs in order to preserve injectivity, and hence expressivity. Some of the existing pooling methods utilize the sum aggregation without an MLP as `RED`. These can still maintain and increase expressivity when an MLP is applied before `RED` by another operation, e.g., as the final part of a preceding MPNN layer. For example, a GIN layer applies an MLP as its last operation. In the next section, we discuss conditions on `SEL` that lead to a pooling operator that not only maintains but increases expressive power.

## 4 Conditions for Increasing the Expressive Power

**Definition 4.1** (Increasing Expressivity). A pooling operator `POOL = (SEL, RED, CON)` is **increasing expressivity** if it is maintaining expressivity and if there is a pair of graphs that are WL indistinguishable, which become WL-distinguishable after pooling, i.e., if there exist $\mathcal{G}_1 =_{WL} \mathcal{G}_2$ with $\texttt{POOL}(\mathcal{G}_1) \neq_{WL} \texttt{POOL}(\mathcal{G}_2)$.

In what follows, we assume that the `RED` operator is M-`RED` as in Proposition 3.3. Whenever we can obtain different supernode assignments for two graphs, an injective M-`RED` will correspondingly construct different aggregated features. Finding such assignments for some pair of WL indistinguishable graphs allows us to construct a pooling method that distinguishes those graphs. Combining this insight with Proposition 3.2, we also know that all cases distinguished by WL on the original graphs can remain distinguishable after pooling the graphs. Thus, to achieve a strictly increased expressivity of a model utilizing pooling, it is sufficient to use M-`RED` and to select a suitable `SEL` function that can assign two WL-indistinguishable graphs to different supernode assignments. Notably, the `SEL` operation does not need to be more expressive than WL; mere incomparability suffices. We formalize this in the following statement.

**Theorem 4.2.** *Let* `POOL` *be a pooling operator expressed by the functions* `SEL`, *M-*`RED`, *CON*, *where* *SEL* *is incomparable to or more expressive than the WL test, then* `POOL` *increases the expressive power and is at least as expressive as* *SEL*.

*Proof.* By Proposition 3.3, `POOL` maintains the expressivity; thus, in order to prove that `POOL` increases expressivity we need to find two graphs $\mathcal{G}_1 =_{WL} \mathcal{G}_2$ such that the corresponding pooled graphs $\texttt{POOL}(\mathcal{G}_1) = (\mathcal{V}_1^P, \mathcal{E}_1^P, \mathbf{H}_1^P)$ and $\texttt{POOL}(\mathcal{G}_2) = (\mathcal{V}_2^P, \mathcal{E}_2^P, \mathbf{H}_2^P)$ are WL distinguishable. By hypothesis, `SEL` is incomparable to or more expressive than the WL test. By Definition 2.4, in both cases, there exist graphs $\mathcal{G}_1$, $\mathcal{G}_2$ with $\mathcal{G}_1 =_{WL} \mathcal{G}_2$ and $\texttt{SEL}(\mathcal{G}_1) \neq_{WL} \texttt{SEL}(\mathcal{G}_2)$. By Definition 2.6 of `SEL` this implies

$$\{\!\{\{\!\{ s_v^j \ : \ v \in \mathcal{V}_1 \}\!\} \ : \ j \in \mathcal{V}_1^P \}\!\} \neq \{\!\{\{\!\{ s_v^j \ : \ v \in \mathcal{V}_2 \}\!\} \ : \ j \in \mathcal{V}_2^P \}\!\}$$

We can make a case distinction:

**There exists a supernode $j \in \mathcal{V}_1^P$ such that:**

$$\{\!\{ s_v^j \ : \ v \in \mathcal{V}_1 \}\!\} \neq \{\!\{ s_v^i \ : \ v \in \mathcal{V}_2 \}\!\} \ \forall i \in \mathcal{V}_2^P. \tag{3}$$

Since $\mathcal{G}_1 =_{WL} \mathcal{G}_2$, then $|\mathcal{V}_1| = |\mathcal{V}_2|$, so Equation (3) implies that there exists an $\mathbf{s}$ for which the multiplicities are different, i.e.,

$$\left| \{\!\{ (\mathbf{h}_v, s_v^j) \ : \ v \in \mathcal{V}_1 \wedge s_v^j = \mathbf{s} \}\!\} \right| = k \neq l = \left| \{\!\{ (\mathbf{h}_v, s_v^i) \ : \ v \in \mathcal{V}_2 \wedge s_v^j = \mathbf{s} \}\!\} \right| \ \forall i \in \mathcal{V}_2^P. \tag{4}$$

This implies

$$\{\!\{\{\!\{ (\mathbf{h}_v, s_v^j) \ : \ v \in \mathcal{V}_1 \}\!\} \ : \ j \in \mathcal{V}_1^P \}\!\} \neq \{\!\{\{\!\{ (\mathbf{h}_v, s_v^j) \ : \ v \in \mathcal{V}_2 \}\!\} \ : \ j \in \mathcal{V}_2^P \}\!\}$$

which due to injectivity of M-`RED` (Proposition 3.2 and Proposition 3.3) and the use of `SEL` that is incomparable or more expressive than WL implies that $\mathcal{G}_1^P \neq_{CD} \mathcal{G}_2^P$.

**The two pooled graphs have different number of supernodes, i.e.,**

$$\|\{\!\{\{\!\{ s_v^j \ : \ v \in \mathcal{V}_1 \}\!\} \ : \ j \in \mathcal{V}_1^P \}\!\}\| \neq \|\{\!\{\{\!\{ s_v^j \ : \ v \in \mathcal{V}_2 \}\!\} \ : \ j \in \mathcal{V}_2^P \}\!\}\|$$

which implies $\mathcal{G}_1^P \neq_{CD} \mathcal{G}_2^P$.

As this proof holds for any $\mathcal{G}_1, \mathcal{G}_2$ with $\mathcal{G}_1 =_{WL} \mathcal{G}_2$ and $\texttt{SEL}(\mathcal{G}_1) \neq_{WL} \texttt{SEL}(\mathcal{G}_2)$, `POOL` is also at least as expressive as `SEL`. □

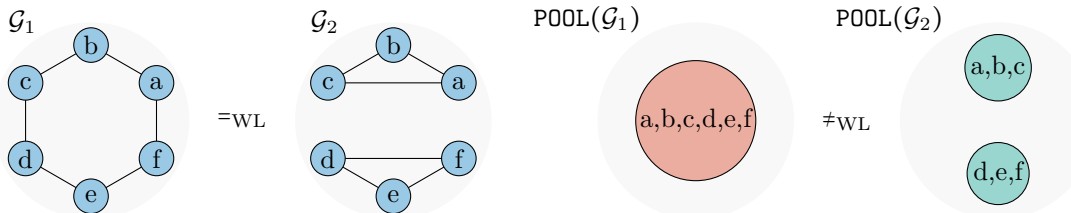

SEL: Cluster connected vertices together

Figure 2: Two WL-indistinguishable graphs $\mathcal{G}_1, \mathcal{G}_2$ which can be distinguished after merging all connected components. This combines the two cycles $\mathcal{G}_2$ to a two supernode graph, while it maps $\mathcal{G}_1$ to a single supernode graph, which are WL distinguishable.

Given this insight, we now have the theoretical confirmation that graph pooling can increase the expressivity of GNNs by utilizing a powerful node selection operator. Note that Theorem 4.2 identifies a lower bound on the expressivity. Setting CON to return no edges at all, a specific choice of SEL implies that the expressivity of POOL = (SEL, M-RED, CON) is bounded by the direct sum of the equivalence relations defined by WL and by SEL. That is, two graphs that differ in either WL colors or the output of SEL, or both, can be distinguished by WL on the pooled graphs, while otherwise they cannot. Adding a nontrivial CON, however, complicates this analysis dramatically, and providing an upper bound is an open question.

As an example of a powerful SEL, consider a function that clusters together nodes that lie on the same cycle as visualized in Figure 2. This SEL function is incomparable to WL, as it cannot distinguish non-isomorphic trees of the same size. Figure 2 shows that two triangles and a cycle of length six become WL-distinguishable after pooling with such a SEL function, although these graphs are WL-indistinguishable before pooling. Several existing methods utilize SEL functions that are incomparable to WL, such as CliquePool (Luzhnica et al., 2019) selecting based on cliques and CurvPool (Sanders et al., 2023) selecting based on graph curvature. It is worth noting that these methods utilize additional time-consuming graph algorithms to select nodes and are not able to take node representations into account. For instance, CliquePool identifies cliques in a given graph using an algorithm by Bron & Kerbosch (1973) which has a worst-case runtime of $\mathcal{O}(3^{n/3})$ for $n$ nodes.

## 4.1 Edge-Based Pooling Improves Expressivity

So far, we have shown that with an injective RED and a powerful SEL, we can enhance expressivity beyond the WL test, without any conditions on the CON function. In the following, we prove how leveraging a suitable CON can further improve expressivity when combined with a SEL operator that exploits the connectivity.

**Definition 4.3** (M-CON). Let SEL: $\mathcal{G} \to \mathbf{S}$ be a select function with $\mathbf{S}_j = (s_i^j)_{v=1}^n$ where $s_v^j$ indicates the contribution of node $v$ on supernode $j$. We define the M-CON function as connecting two supernodes if and only if they contain nodes that were connected in the original graph as follows:

$$\text{M-CON}(\mathcal{G}, \mathbf{S}) = \{(m_1, m_2) \mid s_u^{m_1}, s_v^{m_2} \neq 0, \ (u, v) \in \mathcal{E}\}.$$

Given M-CON as in Definition 4.3 and M-RED, as in Proposition 3.3, we can now show that there are some cases that Theorem 4.2 does not cover where the resulting pooled graphs are distinguishable.

**Theorem 4.4.** *Let POOL = (SEL, M-RED, M-CON) with M-CON as in Definition 4.3 and M-RED, as in Proposition 3.3. Then, there exists a SEL function and two graphs $\mathcal{G}_1 =_{WL} \mathcal{G}_2$ such that SEL($\mathcal{G}_1$) = SEL($\mathcal{G}_2$) but POOL($\mathcal{G}_1$) $\neq_{WL}$ POOL($\mathcal{G}_2$).*

*Proof.* Let $\mathcal{G}_1$ be composed of two triangles connected by an edge. Let $\mathcal{G}_2$, be a hexagon with an additional edge between two nodes with distance three. We have $\mathcal{G}_1 =_{WL} \mathcal{G}_2$. This scenario is visualized in Figure 3. Now, let SEL be constructed such that for all adjacent pairs of nodes $u, v \in \mathcal{V}$ with degree two, we create a

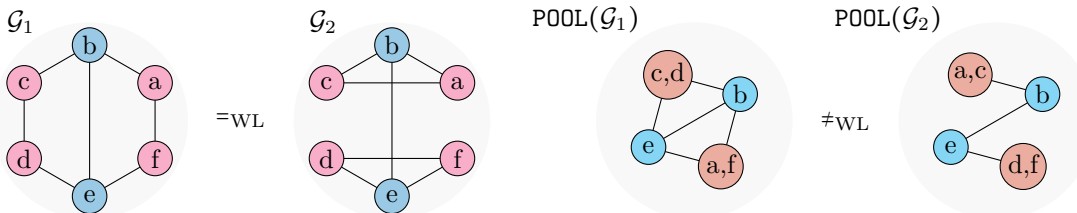

SEL: Cluster connected {⬤⬤} together

CON: Add an edge between supernodes if constituents were connected

Figure 3: Two WL-indistinguishable graphs $\mathcal{G}_1$, $\mathcal{G}_2$ which can be distinguished after pooling as in the proof of Theorem 4.4. Supernodes correspond to connected components of subgraphs induced by individual node colors. Supernodes $a, b$ are connected by an edge if and only if there is an edge $v, w$ in the original graph where $v$ is pooled into $a$ and $w$ is pooled into $b$. Here, this results in pooled graphs with different numbers of edges, which are WL distinguishable.

supernode $j$ and set $s_v^j \neq 0$ and $s_u^j \neq 0$. Using the aforementioned construction of SEL, we observe (cf. Figure 3 on the right) that $\mathtt{SEL}(\mathcal{G}_1) = \mathtt{SEL}(\mathcal{G}_2)$ while $\mathtt{POOL}(\mathcal{G}_1)$ and $\mathtt{POOL}(\mathcal{G}_2)$ have a different number of edges. Thus, $\mathtt{POOL}(\mathcal{G}_1) \neq_{WL} \mathtt{POOL}(\mathcal{G}_2)$. □

In the proof, we identified two graphs that could not be distinguished using a pooling function defined as in Theorem 4.2. However, they can be distinguished when a specific CON, that is, M-CON, is applied. Therefore, employing M-CON function makes the pooling mechanism more expressive. The proof emphasizes the significance of including the node connectivities when clustering to supernodes. While the pooled multisets of node representations may still be equal, the graph's structure can differ. This would allow message-passing to distinguish two graphs as we consider each node's neighborhood. Therefore, pooling operators are more powerful regarding their separation ability of graphs if the selection is based on the node's representations and the graph's topology, cf. Section 4.2. Consequently, pooling methods that consider topological information when forming supernodes are desired. Appendix A.1 shows further examples of SEL functions that increase expressivity when combined with M-RED and M-CON. Forming a supernode from $k$ nodes is similar to $k$-WL, as the graph resulting from $k$-WL equivalently contains a supernode formed by these nodes. However, as the expressivity depends on CON, we cannot provide a general upper bound based on the WL hierarchy.

## 4.2 Node-Based Pooling Cannot Improve Expressivity

In this section, we show that pooling methods, taking just the node representations or topological information that do not go beyond WL for SEL, cannot increase expressivity when combined with M-RED and M-CON. This implies that some practical pooling operators such as DMoN (Tsitsulin et al., 2023) do not increase expressivity. This also applies to methods like DiffPool (Ying et al., 2018), as we show in Section 5.3. We start by defining a generic node-based select operator:

**Definition 4.5.** (Node-based Select) For a given graph $\mathcal{G} = (\mathcal{V}, \mathcal{E}, \mathbf{H})$ where $\mathbf{H}$ is computed by an MPNN, a node-based select operator is defined as

$$\mathrm{NB}\text{--}\mathtt{SEL}(\mathcal{G}) = \mathbf{S} \in \mathbb{R}^{n \times k} \text{ with } \mathbf{S}_j = (s_v^j)_{v=1}^n, s_v^j = g_j(\mathbf{h}_v) \ \forall j \in [1, ..., k] \tag{5}$$

i.e., the contribution functions $g_j \colon \mathbb{R}^d \to \mathbb{R}$ take only MPNN-computed node representations as arguments.

In the following proposition, we show that not considering the edge set as an argument in the SEL function, together with using the M-CON function, makes it impossible to increase expressivity.

**Proposition 4.6.** *Let* POOL $= ($ NB-SEL, M-RED, M-CON$)$, *with* NB-SEL, M-RED, M-CON *as defined in Definitions 4.3 and 4.5 and Proposition 3.3. Then,* POOL *does not increase expressivity.*

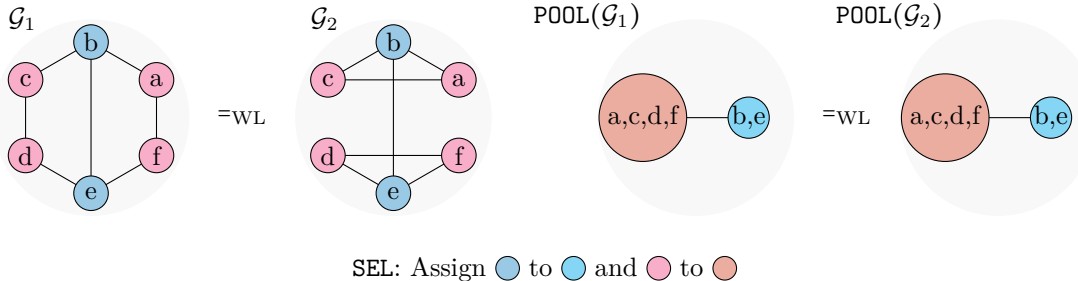

SEL: Assign 🔵 to 🔵 and 🔴 to 🔴

M-CON: Add an edge between supernodes if constituents were connected

Figure 4: Two WL-indistinguishable graphs $\mathcal{G}_1$, $\mathcal{G}_2$ which cannot be distinguished after pooling with a node based SEL function as in Definition 4.5.

*Proof.* Let $\mathcal{G}_1 = (V_1, \mathcal{E}_1, \mathbf{H}_1)$, $\mathcal{G}_2 = (V_2, \mathcal{E}_2, \mathbf{H}_2)$ be two non-isomorphic but WL indistinguishable graphs, i.e. $\mathcal{G}_1 =_{WL} \mathcal{G}_2$. Let $Y_1 = Y_2$ be the labels after convergence of the WL algorithm on $\mathcal{G}_1$ and $\mathcal{G}_2$, respectively. Further, let $\text{POOL}(\mathcal{G}_1) = (\mathcal{V}_1^P, \mathcal{E}_1^P, \mathbf{H}_1^P)$, $\text{POOL}(\mathcal{G}_2) = (\mathcal{V}_2^P, \mathcal{E}_2^P, \mathbf{H}_2^P)$ be the pooled graphs. Since the contribution functions $g_p \colon \mathbb{R}^d \to \mathbb{R}$ for the supernodes take only node representations as arguments, for every $v_1 \in \mathcal{V}_1$ and $v_2 \in \mathcal{V}_2$ with $\mathbf{h}_{v_1} = \mathbf{h}_{v_2}$, we have $s_{v_1}^j = s_{v_2}^k$ for all supernodes $j \in \mathcal{V}_1^P$ and $k \in \mathcal{V}_2^P$. As such the multisets $\{\!\{\{\!\{(\mathbf{h}_v, s_v^j) \colon v \in V_1\}\!\} \colon j \in \mathcal{V}_1^P\}\!\} = \{\!\{\{\!\{(\mathbf{h}_v, s_v^j) \colon v \in V_2\}\!\} \colon j \in \mathcal{V}_2^P\}\!\}$ are equal. By definition of M-RED we have $\mathbf{H}_1^P = \mathbf{H}_2^P$. For every $(v_1, u_1) \in \mathcal{E}_1$ there exists a $v_2 \in \mathcal{V}_2$ with $\mathbf{h}_{v_1} = \mathbf{h}_{v_2}$ with corresponding $u_2 \in \mathcal{N}_{v_2}$ and $\mathbf{h}_{u_1} = \mathbf{h}_{u_2}$. Due to $\mathbf{H}_1^P = \mathbf{H}_2^P$, there also exists a bijection $\psi \colon \mathcal{V}_1^P \to \mathcal{V}_2^P$ between supernodes such that for every supernode $v_2^P = \phi(v_1^P)$ we have $\mathbf{h}_{v_1}^P = \mathbf{h}_{v_2}^P$. For every supernode $v_1^P \in \mathcal{V}_1^P$ and $u_1^P \in \mathcal{V}_1^P$ with $s_{v_1}^{v_1^P} \neq 0$ and $s_{u_1}^{u_1^P} \neq 0$, let $v_2^P = \psi(v_1^P)$ and $u_2^P = \psi(u_1^P)$. For these, we also have $s_{v_2}^{v_2^P} \neq 0$ and $s_{u_2}^{u_2^P} \neq 0$ as $g_j$ takes only the equal node representations as arguments. Thus, for every pair of connected supernodes in $\mathcal{V}_1^P$ there exists a corresponding pair of connected supernode in $\mathcal{V}_2^P$. □

Many existing methods, e.g., MinCutPool (Bianchi et al., 2020a), DMoN (Tsitsulin et al., 2023) or Diff-Pool (Ying et al., 2018) can be directly expressed using a node-based select because they exclusively utilize MPNNs within the SEL operator. When the SEL operator does not use topological information (beyond WL) of the graph but only node representations resulting from message-passing, this is equivalent to performing additional message-passing steps before applying the pooling operator. We provide the formal details on this equivalence in Section 5.3. Figure 4 shows a node-based SEL function that does not increase expressivity.

This example confirms that methods like DiffPool (Ying et al., 2018), MinCutPool (Bianchi et al., 2020a), or DMoN (Tsitsulin et al., 2023) are unable to improve the expressivity of a GNN. Based on Bianchi & Lachi (2023), these methods can be constructed to maintain expressivity. Contrarily, Bianchi & Lachi (2023) have shown that methods like Top-k pooling (Gao & Ji, 2019) and SAGPool (Lee et al., 2019) do not maintain expressivity in certain cases. As these methods can similarly be expressed as node-based pooling, Proposition 4.6 provides the complimentary result that expressivity cannot be improved in any case. Thus, these methods are strictly less expressive than WL.

Our findings in this section show that pooling methods can increase the expressivity of GNNs. We outlined two directions. In Theorem 4.2, we showed that whenever SEL can assign two nodes with identical WL-colors to different supernodes, this leads to an increase in expressivity of the GNN after pooling. This is the case when SEL is more expressive than WL or incomparable to WL. The pooling then increases the expressivity as resulting equal graphs are combined with discriminative supernode representations by M-RED. Based on Theorem 4.4, the second option to increase expressivity is to include the connectivity of node representations. This does not require a complex SEL function as it suffices to combine only connected nodes based on given node representations. Contrarily, when selecting node representations without considering topological information beyond WL, expressivity cannot be increased, as shown in Proposition 4.6.

# 5 Systematization of Pooling Operators

This section describes how existing pooling methods align with our theoretical analysis. To further illustrate our theoretical insights, we introduce XP — a pooling strategy that contracts a single edge per graph. XP is not intended as a state-of-the-art method, but rather as an example showing how even simple operations can satisfy the expressivity-increasing criteria derived from the theory. It demonstrates that increasing expressivity can be achieved with minimal structural intervention and serves as proof of concept among many possible constructions.

## 5.1 Expressive `SEL` functions (Theorem 4.2)

We describe several existing methods capable of obtaining different supernode assignments for two WL indistinguishable graphs by utilizing a powerful node select function. This corresponds to Theorem 4.2. These methods utilize M-`CON` as defined in Definition 4.3.

**CliquePool** CliquePool (Luzhnica et al., 2019) aggregates nodes into maximal cliques, ensuring that each supernode represents a set of fully connected nodes. Using the Bron-Kerbosch algorithm, maximal cliques are detected and prioritized by size. Each node is greedily assigned to the largest unassigned clique, and pooled features are computed via average or max operations. As the WL test cannot identify cliques in a graph, CliquePool is increasing expressivity. However, the Bron-Kerbosch algorithm for a graph with $n$ nodes has a worst-case runtime of $\mathcal{O}(3^{n/3})$.

**CurvPool** In CurvPool (Sanders et al., 2023), sets of nodes are clustered into a supernode when these are connected by a high or a low graph curvature. The considered Balanced Forman curvature (BFC) indicates whether the two adjacent nodes are densely or sparsely connected for each edge. As the BFC can be different for WL indistinguishable graphs (Topping et al., 2022), CurvPool can increase expressivity. While clustering densely connected sets of nodes is similar to CliquePool, computing the BFC has a significantly reduced complexity of $\mathcal{O}(|\mathcal{E}|d_{\max}^2)$ where $d_{\max}$ is the maximum node degree.

**ASAPool** ASAPool (Ranjan et al., 2020) first identifies all potential local clusters within a given graph. A self-attention mechanism is employed to learn a cluster assignment matrix for each cluster. To do so, a new self-attention variant, Master2Token (M2T), is introduced. The clusters are evaluated using an MPNN, and the highest-scoring clusters are selected to create the nodes of the pooled graph. By only allowing supernodes to be formed within local clusters, ASAPool can obtain different supernode assignments for WL indistinguishable graphs as these can form different clusters. However, not all node representations contribute in the `RED` operation, resulting in potentially decreasing expressivity, as shown by Bianchi & Lachi (2023). ASAPool has linear runtime complexity of $\mathcal{O}(n + |\mathcal{E}|)$.

## 5.2 Edge-based Pooling (Theorem 4.4)

Based on Theorem 4.4, expressivity-increasing pooling operators can also be constructed by selecting edges for which the adjacent nodes are combined into supernodes. While the aforementioned pooling operators based on an expressive `SEL` may also be categorized as edge-based, we group methods that closely follow the scheme described in Theorem 4.4 here. We introduce a novel pooling operator that closely follows our theory and describe how two existing pooling operators already follow similar ideas without being explicitly designed to increase expressivity.

**EXpressive Pooling (XP)** By following the constructive proof of Theorem 4.4 in combination with M-`CON`, from Definition 4.3, we obtain the expressivity-increasing pooling operator XP. For each edge, we calculate an edge score $p_{v,u}$ utilizing an MLP on the representations $\mathbf{h}_v$, $\mathbf{h}_u$ of adjacent nodes $v, u$. The two nodes adjacent to the edge with the highest score are combined into a single supernode, while all other nodes form their own supernode. This corresponds to a minimal loss of information about the graph structure. Note that in the case of repeated maximal edge scores, their ordering and the corresponding node selection may be non-deterministic, and a graph can be pooled in multiple ways. Also, note that this particular edge selection is not differentiable but rather serves as a proof of concept. The new graph is constructed as given by M-`CON` in Definition 4.3. Thus, two supernodes are connected if and only if they result from nodes

connected in the original graph. To assure that expressivity is maintained in all cases, we approximate the expressivity-maintaining reduce operator `M-RED` given by Proposition 3.3 using an MLP. This process has runtime complexity $\mathcal{O}(n + |\mathcal{E}|)$.

**ClusterPool** Similarly to XP, ClusterPool (Snelleman et al., 2024) calculates edge scores based on the adjacent node representations. However, all edges above a tunable threshold are selected. Based on the selected edges, each connected component is merged into a single supernode and nodes for which none of the adjacent edges is selected form their own supernode. While XP is a special case of ClusterPool, ClusterPool may significantly reduce the size of the graph, which can cause information to be lost. The runtime complexity is also $\mathcal{O}(n + |\mathcal{E}|)$ (Snelleman et al., 2024).

**EdgePool** The EdgePool (Diehl, 2019) similarly computes a score for each edge. Iteratively, the edge with the highest score is selected, and adjacent nodes are combined into a supernode. Edges for which one of the nodes was already selected are discarded after each iteration. This is repeated until all edges are either selected or discarded. As `SEL` contracts as many edges as possible, much structural information will be lost. EdgePool is non-deterministic. As with XP and ClusterPool, runtime complexity is $\mathcal{O}(n + |\mathcal{E}|)$ (Diehl, 2019).

### 5.3 Node-based Pooling Operators

We now discuss several existing approaches for graph pooling that can be expressed by a node-based select operator. Therefore, these methods cannot increase expressivity.

**DiffPool** DiffPool (Ying et al., 2018) is a differentiable pooling operator that learns a soft cluster assignment for each node to a fixed number $c$ of supernodes. It utilizes an additional MPNN to map the state $\mathbf{X}$ to new node representations $\mathbf{X}'$ and then maps each node representation to assignment probabilities $s_v^j = f_j(\mathbf{x}_v')$ for each supernode $j$. It has a theoretical runtime complexity of $\mathcal{O}(n^2)$. While the supernode assignment considers the graph topology, we now show that it can be equivalently expressed as a node-based pooling operator and that the topological information do not go beyond WL and therefore DiffPool cannot increase expressivity:

**Example 5.1** (DiffPool is node-based)**.** Given are a graph $\mathcal{G} = (\mathcal{V}, \mathcal{E}, \mathbf{X})$ and hidden node representations $\mathbf{H} = f(\mathcal{G})$ obtained by message-passing layers $\psi$. DiffPool utilizes additional message-passing layers $g_1$ and the node-wise softmax operation within the `SEL` operation to obtain node assignment scores

$$\mathbf{S} = \text{softmax}(g_1((\mathcal{V}, \mathcal{E}, \mathbf{H})))$$

based on the graph topology. The `RED` function obtains the new node representations by multiplying the supernode assignment scores $\mathbf{S}$ with $\mathbf{H}$, on which another set of message-passing layers $g_2$ are applied, i.e.,

$$\mathbf{H}' = \mathbf{S} \cdot g_2((\mathcal{V}, \mathcal{E}, \mathbf{H})) .$$

These additional message-passing operations can be factored out of the pooling operation by redefining the initial $\psi$ as

$$\hat{\psi}(\mathcal{G}) := (\mathbf{H}_1, \mathbf{H}_2) = (g_1((\mathcal{V}, \mathcal{E}, \psi(\mathcal{G}))), g_2((\mathcal{V}, \mathcal{E}, \psi(\mathcal{G})))) .$$

Consequently, the supernode assignment scores and updated node representations can then equivalently be obtained without using the graph topology as a node-based pooling by

$$\mathbf{S} = \text{softmax}(\mathbf{H}_1), \quad \mathbf{H}' = \mathbf{S}\mathbf{H}_2.$$

**Top-k** The Top-k pooling operator (Gao & Ji, 2019) starts with assigning scores to each node. The $k$ nodes with the highest score, identified as the most informative, are retained, while the rest are discarded. As the node scoring is based solely on node features, Top-k pooling is a node-based select method. The runtime complexity of Top-k pooling is $\mathcal{O}(n + |\mathcal{E}|)$ (Gao & Ji, 2019).

**SAGPool** SAGPool (Lee et al., 2019) employs a self-attention mechanism with graph convolution to effectively discern which nodes should be retained or dropped. Unlike Top-k, the self-attention mechanism

is based on graph convolutions which enables SAGPool to consider both node features and graph topology. However, when the graph's topology is considered through message-passing layers, SAGPool can be stated equivalently as a node-based pooling operator similar to DiffPool (Example 5.1). However, SAGPool has an improved runtime complexity of $\mathcal{O}(n + |\mathcal{E}|)$ (Lee et al., 2019).

# 6 Experiments

To validate our theoretical findings and evaluate the expressivity-increasing capabilities of various pooling operators, we compare their empirical expressivity using the BREC framework (Wang & Zhang, 2024). Our reproducible implementation is publicly available [1].

## 6.1 BREC Framework

We evaluate the empirical expressivity of all considered methods using the BREC framework (Wang & Zhang, 2024). The BREC dataset comprises 400 pairs of non-isomorphic, WL-indistinguishable graphs, which allows for an empirical evaluation of various methods in their ability to distinguish those graphs. The graphs are divided into four main categories: Basic, Regular, Extension, and CFI graphs. Basic Graphs include 60 pairs of 10-node graphs, Regular Graphs consist of 140 pairs, including simple and strongly regular graphs, Extension Graphs are made up of 100 pairs inspired by GNN extensions, and CFI Graphs, the most complex category includes 100 pairs based on the Cai-Fürer-Immerman method (Cai et al., 1992). 270 of these graphs are 3-WL distinguishable. The BREC dataset stands out for its range of graph difficulties, different assessments of GNNs, and large scale, making it a significant benchmark for studies in GNN expressivity.

The framework takes pairs of WL indistinguishable graphs, creates multiple permutations of both graphs and optimizes the parameters of a given GNN using contrastive learning to distinguish these. The framework computes a distance score between embeddings and counts each pair as distinguished if the score is larger than a predefined threshold. If a pair is distinguished, the framework also requires embeddings of additional permutations of the same graph to be sufficiently close. Otherwise, the sample is considered to be unreliable, and the total score will not be increased.

## 6.2 Evaluated Methods

All hierarchical pooling architectures use a uniform structure to ensure consistency across different implementations. The initial feature of each node is determined by its degree. We apply a total of twelve MPNN layers, with one pooling layer added after four and after eight MPNN layers. Global sum pooling, followed by a two-layer MLP, is used to produce graph-level embeddings that are optimized to differ between non-isomorphic graphs. As MPNN layers, we used the GIN layer (Xu et al., 2019) due to its ability to match the expressivity of the WL test. As pooling layers, we evaluate all methods described in Section 5, which can be divided into node-based operators, edge-based operators, and operators using an expressive SEL function.

## 6.3 Experimental Results

The maximum scores over ten runs, as proposed by Wang & Zhang (2024) are displayed in Table 1. Additionally, we show the average runtime in seconds for evaluating the BREC framework for a single seed. We provide implementation details and more fine-grained results in Appendix A.

The results confirm our theoretical findings. Node-based pooling approaches fail to enhance expressivity, as evidenced by their inability to distinguish any graph structures across all tested scenarios. This aligns with our theoretical predictions, given that these methods primarily operate at the node level without significantly leveraging the graph topology. In contrast, edge-based methods successfully improve expressivity, with XP demonstrating the highest performance. This can be attributed to its minimal structural modifications, which may preserve more relational information. As ClusterPool requires a fixed threshold, we observed most edge scores to be either below or above this threshold. We expect results to be more similar when modifying ClusterPool to only the maximal edge scores.

---

[1] https://github.com/aliicee3/brec-pooling

Table 1: Number of graph pairs from the BREC dataset correctly distinguished by different pooling operators in combination with GIN message-passing. Maximum scores over ten runs, as proposed by Wang & Zhang (2024). For comparison, we report results for some higher-order GNNs taken from Wang & Zhang (2024). However, their runtimes were obtained on different hardware and are not directly comparable with our runtimes.

| Pooling Methods | | Basic (60) | Regular (140) | Extension (100) | CFI (100) | Total (400) | Runtime (s) |
|---|---|---|---|---|---|---|---|
| | Global Sum Pooling | 0 | 0 | 0 | 0 | 0 | 227 |
| Node-based Proposition 4.6 | DiffPool | 0 | 0 | 0 | 0 | 0 | 330 |
| | Top-k | 0 | 0 | 0 | 0 | 0 | 267 |
| | SAGPool | 0 | 0 | 0 | 0 | 0 | 290 |
| Edge-based Theorem 4.4 | EdgePool | 13 | 0 | 10 | 0 | 23 | 1539 |
| | ClusterPool | 12 | 00 | 15 | 0 | 27 | 2703 |
| | XP | 48 | 3 | 72 | 1 | 124 | 303 |
| Expressive SEL Theorem 4.2 | CliquePool | 44 | 46 | 32 | 0 | 122 | 2579 |
| | CurvPool | 35 | 65 | 53 | 4 | 157 | 11657 |
| | ASAPool | 2 | 1 | 8 | 2 | 13 | 1290 |
| Higher Order GNNs (Wang & Zhang, 2024) | | | | | | | |
| | PPGN | 60 | 50 | 100 | 23 | 233 | 477[+] |
| | GSN | 60 | 99 | 95 | 0 | 254 | >5381[+] |
| | DSS-GNN | 58 | 48 | 100 | 15 | 221 | 3351[+] |
| | $\delta$-$k$-LGNN | 60 | 50 | 100 | 6 | 216 | 2537[+] |
| | Graphormer | 16 | 12 | 41 | 10 | 79 | >>13747[+] |

Expressive SEL methods increase expressivity, particularly excelling in distinguishing regular graphs due to their ability to detect triangles. ASAPool, despite leveraging a powerful SEL mechanism, does not achieve good results because, as explained in Section 5.2, the RED function does not guarantee the preservation of expressivity. Despite the good performance of CliquePool and CurvePool in graph distinction, these methods exhibit the longest runtime. Conversely, XP achieves comparable graph differentiation performance while maintaining a low runtime, on par with node-based methods. This balance between expressivity and computational efficiency makes XP a compelling choice for practical scenarios where both expressivity and execution time are critical factors.

For comparison, we also include expressivity results for higher-order GNNs reported in Wang & Zhang (2024), namely PPGN (Maron et al., 2019), GSN (Bouritsas et al., 2023), DSS-GNN (Bevilacqua et al., 2022), $\delta$-$k$-LGNN (Morris et al., 2020), and Graphormer (Ying et al., 2021). We also report the combined runtime of preprocessing and evaluation reported in Table 11 of Wang & Zhang (2024). Note that the runtimes were obtained on different hardware (indicated by [+] in Table 1) and that GSN and Graphormer runtimes only include a subset of the data. We observe that most higher-order methods achieve near-perfect separation on both Basic and Extension graphs, and they consistently outperform the pooling methods on all subsets. While the runtimes are not directly comparable, the higher-order GNNs have a much larger theoretical runtime complexity. PPGN, as the fastest higher-order GNN, has a cubic runtime complexity, whereas the edge-based pooling methods have a linear runtime complexity.

# 7 Conclusion

In this work, we identified two sufficient conditions for pooling operators to increase the expressivity of MPNNs while reducing the size of the computational graph. Pooling operators can either utilize sufficiently powerful select operators or the locality of representations by only selecting connected nodes. While, until now, methods that increase the expressivity beyond the limit imposed by WL were based on modifications of message-passing – typically incurring substantial computational costs – this work represents the first attempt to increase the expressive power of GNNs by exploiting the pooling mechanism. We also proved that pooling methods based only on node representations cannot increase expressivity.

We propose examples of expressivity-increasing pooling operators that reduce the size of the pooled graphs. It follows that the computational complexity of MPNN training and inference on the resulting graph is bounded by the complexity of the operations on the original graph if the architecture is not changed. The specific methods that we propose as examples are almost all linear time. This implies that the computational complexity of building and training the pooled architecture is bounded by the computational complexity of building and training the original architecture. If one employs more expensive select operators (e.g. identifying triangles), the computational complexity may slightly increase.

Additionally, we categorized the most common pooling operators from the literature into three distinct categories based on their contribution to expressivity, i.e. operators that increase expressivity through an expressive SEL function, edge-based pooling methods leveraging graph connectivity to enhance discrimination and node-based pooling methods, which do not increase expressivity due to their limitation of relying solely on node representations. Our empirical evaluation confirmed our theoretical findings, demonstrating that various pooling operators can increase the expressivity of GNNs by varying degrees. Our conditions for increasing expressivity and categorizing into the different expressivity-increasing groups offer a systematic guideline for developing future pooling strategies.

The main limitation of our work is that the benefits of the identified increase in expressivity need to be confirmed for practical purposes. However, as our theory can be applied to many different pooling operators, we leave it open for future work to find the best-performing pooling operator for real-world datasets.

## Acknowledgments

PWs work has been partially funded by the Vienna Science and Technology Fund (WWTF) and the City of Vienna [Grant ID: 10.47379/ ICT22059] (StruDL). AMs work has been partially funded by the Ministry of Education and Research Germany (BMBF), under the funding code 01IS20047A, according to the 'Policy for the funding of female junior researchers in Artificial Intelligence'. ARs work is funded by the Federal Ministry of Education and Research of Germany under grant no. 01IS22094E WEST-AI.

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

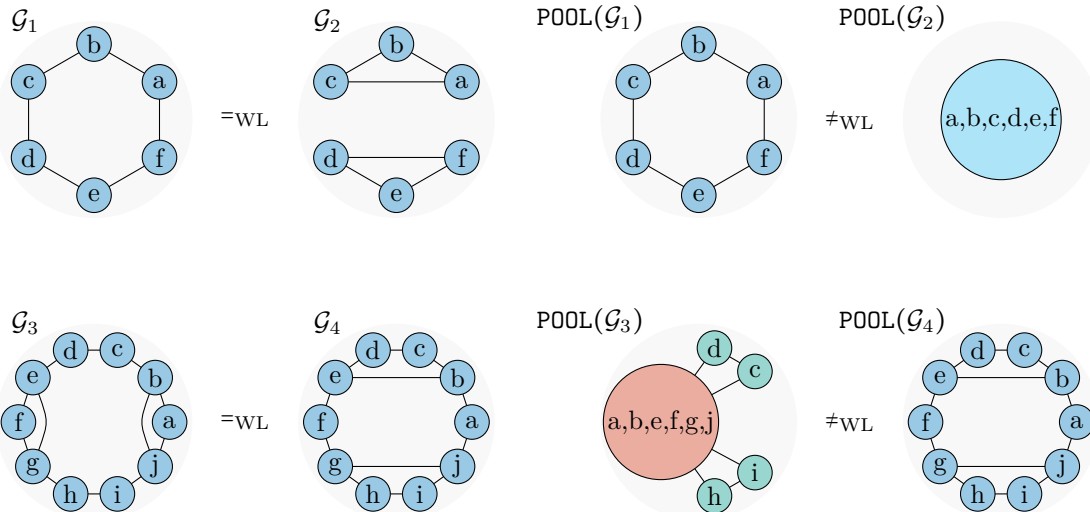

SEL: Cluster vertices on a triangle, retain other vertices

CON: Add an edge between supernodes if constituents were connected

Figure 5: Two pairs of WL-indistinguishable graphs $\mathcal{G}_1, \mathcal{G}_2$ which can be distinguished after clustering all nodes on triangles together.

# A   Appendix

This appendix provides additional (training) details and an ablation study on the interplay of the number of message-passing or pooling layers of our XPpooling and GIN.

## A.1   Further SEL Examples

**Example A.1.** Any POOL method retaining disconnected components when repeatedly applied until all edges are contracted is increasing expressivity.

*Proof.* The ability to maintain expressivity for all graphs follows from Proposition 3.2. In addition, any such POOL method maps two disconnected triangles to two nodes, while the hexagon is mapped to a single node. This is visualized in Figure 2. □

It may seem that expressivity-increasing pooling operators exclusively leverage connectivity by clustering (some) connected nodes while keeping different connected components in separate supernodes. However, this is not a necessary condition to increase expressivity. Figure 5 shows an example pooling operator in which all nodes contained in triangles are clustered into a single supernode. In contrast, nodes that are not on triangles are retained. Other expressivity-increasing pooling operators can be obtained, e.g., by clustering biconnected components into supernodes, as shown in Figure 6. Necessary for increasing expressivity, at least for our choice of the CON function, seems to be the incorporation of the edge set in some form.

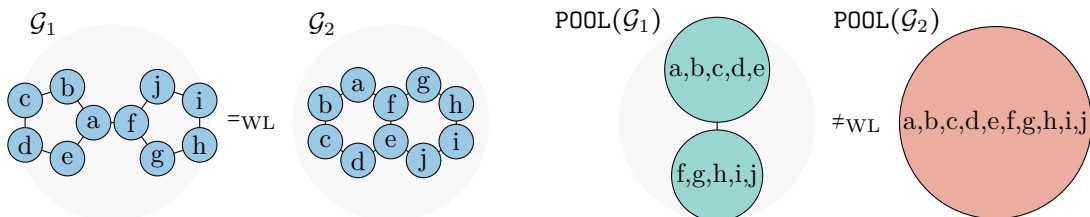

SEL: Cluster vertices in the same biconnected component

CON: Add an edge between supernodes if constituents were connected

Figure 6: Two WL-indistinguishable graphs $\mathcal{G}_1, \mathcal{G}_2$ that become WL-distinguishable after pooling, where nodes in the same biconnected component are clustered into the same supernode. A biconnected component is a maximal subgraph where any two nodes lie on a cycle.

## A.2 Training Details

Our implementation is based on the BREC framework (Wang & Zhang, 2024) and PyTorch Geometric (Fey & Lenssen, 2019). All experiments were run in parallel, with each instance on a Platinum 8468 CPU and 5 GB RAM, apart from the experiments with ASAPool, as those required 100 GB RAM. All processes combined take 57 hours. We repeated all experiments ten times with different random seeds. The source code is reproducable and available at https://github.com/aliicee3/brec-pooling..

## A.3 Ablation Study

In the following, we evaluate how different numbers of message-passing iterations and pooling steps affect the empirical gains in expressivity. As in Table 1, GIN layers are used for message-passing and the BREC dataset is considered. We apply $k$ GIN layers followed by one pooling step, and repeat this process $l$ times. After these steps, another $k$ GIN layers are employed. Thus, the model consists of a total of $l$ pooling steps and $k \cdot (l + 1)$ GIN instantiations. Global sum pooling, followed by a two-layer MLP, produces graph-level representations.

Results for XPare presented in Table 2. These results indicate that expressive power mostly improves with more GIN layers and pooling steps. This matches our theory as expressivity can always be maintained. Further, we observe a trade-off between the increase in expressivity and stability regarding the number of message-passing and pooling layers. Keeping the number of one of these two layer types fixed and

Table 2: Number of graph pairs in the BREC dataset correctly distinguished by XP combined with GIN layers. The model consists of $k$ GIN layers and one pooling step, repeated $l$ times. Mean and standard deviation over three runs are shown. Best result in **bold**. NAN refers to numerical runtime errors.

| XP | | 1 | 2 | 3 | 4 | 5 | 6 | 7 | 8 | 9 |
|---|---|---|---|---|---|---|---|---|---|---|
| | | \multicolumn{9}{c}{$k$ GIN layers between pooling layers} | | | | | | | |
| | 1 | $10 \pm 1$ | $26 \pm 4$ | $61 \pm 4$ | $85 \pm 3$ | $92 \pm 3$ | $97 \pm 3$ | $99 \pm 6$ | $90 \pm 9$ | $94 \pm 2$ |
| | 2 | $37 \pm 7$ | $98 \pm 2$ | $109 \pm 6$ | $112 \pm 6$ | $114 \pm 3$ | $114 \pm 6$ | $114 \pm 4$ | NAN | NAN |
| | 3 | $82 \pm 10$ | $113 \pm 2$ | $110 \pm 4$ | $118 \pm 2$ | $117 \pm 2$ | NAN | NAN | NAN | NAN |
| | 4 | $93 \pm 8$ | $112 \pm 2$ | $113 \pm 3$ | $\mathbf{123} \pm 3$ | NAN | NAN | NAN | NAN | NAN |
| $l$ pooling layers | 5 | $96 \pm 3$ | $114 \pm 9$ | $119 \pm 3$ | NAN | NAN | NAN | NAN | NAN | NAN |
| | 6 | $102 \pm 5$ | $121 \pm 5$ | $119 \pm 9$ | NAN | NAN | NAN | NAN | NAN | NAN |
| | 7 | $95 \pm 4$ | $116 \pm 1$ | NAN | NAN | NAN | NAN | NAN | NAN | NAN |
| | 8 | $96 \pm 4$ | $116 \pm 7$ | NAN | NAN | NAN | NAN | NAN | NAN | NAN |
| | 9 | $90 \pm 6$ | $113 \pm 4$ | NAN | NAN | NAN | NAN | NAN | NAN | NAN |

increasing the other first comes with a notable improvement while adding further layers leads to more or less stable results. However, the deeper the network gets, the more frequent numerical errors become. These observations suggest that finding an optimal number of message-passing and pooling layers is key to reliable performance. As the stability of deep models is a general issue, we refer to studying the connections between the expressivity of deep models and vanishing or exploding gradients and over-smoothing to future work.

