# OpenReview forum: "Expressive Pooling for Graph Neural Networks"
_TMLR — Accepted by TMLR_

### Review · Reviewer_aN64 · 2025-03-16

**Summary Of Contributions:**

1. Theoretical Analysis: The paper presents a well-structured theoretical framework, offering conditions under which pooling operators can improve expressivity. This contributes to the understanding of the interplay between pooling and expressivity in GNNs.


2. New Pooling Method (XP): The introduction of XP as a theoretically motivated pooling method is a valuable contribution. XP aligns with the provided theoretical conditions and demonstrates improved expressivity on synthetic benchmarks.

**Audience:**

Yes

**Claims And Evidence:**

No

**Requested Changes:**

The experimental part is very insufficient, and I found little description of the benchmark dataset BREC. How large is this dataset? What is the sota architecture or method in this dataset? What is the significance of achieving certain metrics in this dataset? etc. Besides, I found little implementation details in the appendix.

- Demonstrate Practical Benefits

Show how expressivity improvements translate to better performance in real-world graph learning tasks,  such as OGB and zinc. Otherwise, this research lacks adequate experimentation.

- Compare with Stronger Baselines

Evaluate against higher-order GNNs or subgraph-based models rather than just pooling methods. Besides, why is the vanilla sum/mean pooling not compared?

- Scalability Analysis

Provide complexity analysis and empirical results on large-scale datasets. Besides, what is the impact of some hyperparameters over the model performance, such as pool ratio and GNN layers?

- Clarify Theoretical Novelty

 Better differentiate contributions from prior work in graph expressivity and pooling theory. Specifically, [A].

[A] The expressive power of pooling in Graph Neural Networks. NIPS 2023.

**Strengths And Weaknesses:**

Despite the contributions, there are several crucial concerns:

- Lack of Novelty in Theoretical Contributions

While the authors provide a formalization of expressivity-enhancing pooling, the results largely extend known principles from the WL hierarchy and graph transformations. The proposed expressivity conditions seem incremental rather than groundbreaking.

- Overemphasis on Synthetic Benchmarks

The evaluation is limited to distinguishing WL-indistinguishable graphs using synthetic datasets. The practical benefits of increased expressivity on real-world tasks remain unclear. The paper does not address whether improving expressivity through pooling leads to tangible performance gains in graph classification or regression tasks.

- Limited Practical Impact of XP

While XP achieves improvements in expressivity over baseline pooling methods, its real-world usability is uncertain. The proposed approach lacks rigorous complexity analysis, and it remains unclear how well it scales to large graphs or whether it provides benefits beyond synthetic benchmarks. Besides, it is outperformed by other pooling algorithms in some metrics (e.g., REGULAR, CFI, TOTAL), which failed to convince the audience of the effectiveness of this XP.

- Potential Methodological Weaknesses in Experiments

The paper does not sufficiently discuss hyperparameter tuning or provide a detailed analysis of how results might vary across different GNN architectures beyond GIN. It is unclear whether the findings generalize to more powerful GNNs such as transformer-based graph models.


- Unclear Differentiation from Related Work:

The paper does not sufficiently position itself against prior work in expressive GNN architectures, particularly methods that use subgraph-based approaches to increase expressivity. There is little discussion on whether pooling fundamentally offers advantages over existing higher-order message-passing techniques.

---

> ### Author Response · Authors · 2025-05-06
> **First answer to reviewer aN64**
>
> Dear Reviewer aN64,
>
>
> Thank you very much for your thoughtful and detailed review of our paper. We greatly appreciate your constructive feedback and the valuable suggestions you provided to improve our manuscript.
> We have carefully addressed all of your requested changes in the revised version. The corresponding modifications are highlighted in the PDF.
> Specifically:
>
> > Compare with Stronger Baselines: Evaluate against higher-order GNNs or subgraph-based models rather than just pooling methods. Besides, why is the vanilla sum/mean pooling not compared?
> * We use global sum pooling to get graph-based embeddings (section 6.2). Thus, the result for NO POOLING corresponds to sum pooling.
> * The authors of the BREC benchmark evaluated 23 methods, including higher-order GNNs and subgraph-based models. We refer to (Wang et al, 2024) for their results. In this work, we focused on pooling methods and compared various methods within this direction.
>
> > The experimental part is very insufficient, and I found little description of the benchmark dataset BREC. How large is this dataset? What is the sota architecture or method in this dataset? What is the significance of achieving certain metrics in this dataset? etc. Besides, I found little implementation details in the appendix.
> * In the revised version, in particular in Section 6.1, details about BREC dataset can be found, including its size, structure, and relevance for evaluating expressivity in GNNs. It contains 400 pairs of graphs that cannot be distinguished by WL. The higher the score of a method, the more WL indistinguishable structures it can distinguish.
> * We evaluate all pooling methods using the publicly available implementation of the BREC benchmark. We refer to that paper for all details. Our extension of this implementation is provided as supplementary material. Implementations of all pooling methods are taken from their respective official implementations or PyTorch Geometric. For details on the utilized GNN architecture, we refer to Section 6.2.
> > Demonstrate Practical Benefits: Show how expressivity improvements translate to better performance in real-world graph learning tasks, such as OGB and zinc. Otherwise, this research lacks adequate experimentation.
> * Adding real-world experiments would not strengthen or further validate the theoretical claims of the paper, nor would it alter our conclusions. Recent studies (Morris et al., 2024; Jogl et al., 2024) have highlighted that the relationship between empirical predictive performance and expressivity is weaker than previously assumed. Performance improvements on benchmarks like OGB or ZINC are often attributed to factors such as ease of optimization or favorable inductive biases, rather than genuine gains in expressivity. In this context, we believe that real-world benchmarks are not well suited to evaluating our specific research question. Instead, we employ the BREC dataset, which has been explicitly designed to measure expressivity in a controlled and meaningful way. Apart from XP, all other pooling methods were extensively evaluated on real-world tasks in their respective publications.

---

> ### Author Response · Authors · 2025-05-06
> **Second answer to reviewer aN64**
>
> Additionally,
>
> > Scalability Analysis: Provide complexity analysis and empirical results on large-scale datasets.
> * We added the runtime complexities of all pooling methods to our paper (Section 5), and provide the details here as well. As a baseline, the complexity of k-WL is O(n^(k+1) log(n)) [Immermann & Sengupta, 2019]. The complexity of pooling methods is given by the sum of the complexity of the MPNN and the pooling function. The edge-based pooling (ClusterPool, EdgePool, XP) methods have complexity O(n + e). The node-based pooling methods have a complexity of at least O(n + e). CliquePool has complexity O(3^(n/3)), CurvPool has complexity O(ed^2_{max}). ASAPool has complexity O(n + e). Here, n is the number of nodes, e the number of edges, d_{max} the maximal node degree.
> * While larger improvements in expressivity can be achieved with increased runtime, expressivity can already be improved with linear runtime complexity (edge-based pooling). We see this as a promising direction.
>
> > Besides, what is the impact of some hyperparameters over the model performance, such as pool ratio and GNN layers?
> * We refer to Appendix A.3 for an ablation study on varying the number of GNN layers. Across all methods, we found a smaller number of pooled nodes to be beneficial and have chosen the hyperparameters accordingly.
>
> > Clarify Theoretical Novelty: Better differentiate contributions from prior work in graph expressivity and pooling theory. Specifically, [A] The expressive power of pooling in Graph Neural Networks. NIPS 2023.
> * The paper "The Expressive Power of Pooling in Graph Neural Networks" establishes sufficient conditions under which a pooling layer maintains the expressivity of a GNN, despite reducing the graph structure. In this paper, we go beyond expressivity preservation: we formally show that pooling can be used to increase the expressive power. That is, the graph coarsening step can lead to representations that distinguish graphs that were originally indistinguishable by the base GNN. This is a key conceptual shift—from preserving to enhancing expressivity—and we believe it has important implications for GNN design. We now explain this distinction more clearly in the paper Section 2.
>
>
> We hope that these revisions adequately address your concerns and improve the clarity and completeness of our work. Should you have any additional comments or suggestions, we would be happy to consider them.
>
>
> * Christopher Morris, Fabrizio Frasca, Nadav Dym, Haggai Maron, Ismail Ilkan Ceylan, Ron Levie, Derek Lim, Michael M Bronstein, Martin Grohe, Stefanie Jegelka (2024): Position: Future Directions in the Theory of Graph Machine Learning. ICML
> * Fabian Jogl, Pascal Welke, Thomas Gärtner (2024): Is Expressivity Essential for the Predictive Performance of Graph Neural Networks?. SciForDL@NeurIPS
> * Wang et al. (2024): An Empirical Study of Realized GNN Expressiveness. ICML
> * Neil Immerman, Rik Sengupta (2019): The k-Dimensional Weisfeiler-Leman Algorithm, arxiv:1907.09582.

---

### Review · Reviewer_XapJ · 2025-03-29

**Summary Of Contributions:**

This paper studies the expressive power of pooling layers in graph neural networks. The authors begin by examining the sufficient conditions for pooling layers to preserve expressivity and also propose the conditions under which they can increase expressivity. Then, they conduct an investigation into the expressivity of eight commonly used pooling layers in the literature and introduce a novel pooling method called XP, which aims to increase the expressive power of message passing graph neural networks. Finally, the authors evaluate the performance of XP and several other pooling methods on a synthetic dataset.

**Audience:**

Yes

**Broader Impact Concerns:**

The authors study the expressive power of pooling layers in GNNs, offering insights into the design of GNN architectures. I don’t see any ethical concerns of this work that necessitate a Broader Impact Statement.

**Claims And Evidence:**

Yes

**Requested Changes:**

I would like to suggest the following changes:
- The authors should discuss the theoretical upper bound of the expressivity of pooling layers, rather than merely stating that they increase expressivity.
- The authors should provide justification for proposing XP as a distinct pooling method and explain why it achieves significantly higher performance on the synthetic task.
- The authors should elucidate how XP can be applied to GNNs in practical settings, considering their non-deterministic and non-differentiable nature.
- The authors should validate the performance of various pooling layers on real-world tasks.

A few minor typos:
- There’s an extra “graph” on the second line of the second graph in the Preliminaries section.
- There’s a missing space at the end of the first paragraph following Definition 2.3, between the word “this” and the citation.
- On the first line of the proof of Proposition 3.2, the spacing after “w.r.t.” is incorrect. For guidance on producing the correct spacing after periods in LaTeX, refer to this page: https://tex.stackexchange.com/questions/2229/is-a-period-after-an-abbreviation-the-same-as-an-end-of-sentence-period.

**Strengths And Weaknesses:**

Strengths:
- The authors provide a theoretical framework for pooling layers in GNNs by dividing them into three distinct operations. They derive conditions under which these operations must meet to maintain or increase expressiveness. This framework is highly general and applies to a wide range of commonly used pooling methods in the literature.
- The authors evaluate various pooling methods on a synthetic dataset and demonstrate that their proposed XP achieves both efficiency and comparable performance. These results offer valuable insights for designing graph neural network architectures.
- The paper is well-written, with numerous illustrative examples that simplify the theory.

Weaknesses:
- The authors’ characterization of pooling layers appears to be a special case of higher-order GNNs. Specifically, pooling layers combine nodes into supernodes, apply a reduce operation to generate their features, and then assign edges between the supernodes. This process is quite similar to higher-order GNNs, which also conduct message passing on supernodes. It would be beneficial for the authors to discuss the relationship between their characterization of pooling layers and higher-order GNNs. For instance, it is known that the expressiveness of $k$-GNNs is upper-bounded by $k$-WL. Can we conclude that the expressiveness of pooling layers that selects at most $k$ nodes is also bounded by $k$-WL? If not, where does the expressiveness of pooling layers lie within the WL hierarchy?
- As mentioned in the previous point, the authors only propose conditions that “increase” the expressivity of GNNs, but they do not provide an analysis of the actual expressiveness of GNNs after incorporating such pooling layers.
- The proposed XP appears to be a specific case of ClusterPool, and I don’t perceive the need to introduce it as a distinct pooling method. The authors contend that ClusterPool could potentially reduce the graph size and lead to information loss, but this can be mitigated by selecting an appropriate threshold for edge selection. Consequently, I fail to comprehend why the performance of ClusterPool is significantly lower compared to XP in the experiment, considering that XP is a special case of ClusterPool.
- Furthermore, the proposed XP is non-deterministic, resulting in distinct representations for the same graph. This deviation from the permutation invariant inductive bias of GNNs may have adverse effects on real-world applications. Moreover, since XP is not differentiable, how can it be applied to GNNs for practical tasks? The authors mentioned that they only trained the MLP layers in their experiment. While this is acceptable for evaluating the expressive power of GNNs on a synthetic task, it is impractical to generalize these findings to real-world applications.
- The authors’ experiments are confined to a small synthetic task, and the performance of the pooling layers is not assessed on real-world tasks. The distinction between non-isomorphic graphs and real-world graph representation learning is vastly different. Therefore, the results obtained on a synthetic dataset cannot provide reliable guidance for designing GNNs for practical applications.

---

> ### Author Response · Authors · 2025-05-06
> **First answer to reviewer XapJ**
>
> Dear Reviewer XapJ,
>
>
> Thank you very much for your thoughtful and detailed review of our paper. We greatly appreciate your constructive feedback and the valuable suggestions you provided to improve our manuscript.
> We have carefully addressed all of your requested changes in the revised version. The corresponding modifications are highlighted in the PDF.
> Specifically:
>
> > The authors should discuss the theoretical upper bound of the expressivity of pooling layers, rather than merely stating that they increase expressivity.
> * While Theorem 4.2 identifies that the SEL function is a lower bound on the expressivity, the (trivial) upper bound is given by distinguishing all non-isomorphic graphs. This upper bound can be reached by choosing SEL to distinguish any pair of non-isomorphic graphs (which would result in an inefficient pooling operator).
> * Setting CON to return no edges at all, a specific choice of SEL implies that the expressivity of POOL = (SEL, M-RED, CON) is bounded by the direct sum of the equivalence relations defined by WL and by SEL. That is, two graphs that differ in either WL colors or the output of SEL, or both, can be distinguished by WL on the pooled graphs, while otherwise they cannot. Adding a nontrivial CON, however, complicates this analysis dramatically and we do not have any elegant way, yet, to formalize this. In Theorem 4.4 we discuss the implications of choices for CON, but can currently not provide an upper bound.
> * We added a clarification of this in Theorem 4.2 and a brief discussion on the missing upper bound after the proof.
>
> > Can we conclude that the expressiveness of pooling layers that select at most k nodes is also bounded by k-WL?
> * This would be a great connection, but it is not true in general. While the resulting graph for k-WL also contains a node with these $k$ nodes, the expressivity of POOL also depends on the CON function. Thus, we cannot make this claim in general.
> * We added this discussion after Theorem 4.4.
>
> > The authors should provide justification for proposing XP as a distinct pooling method and explain
> * We want to clarify that XP is not meant to be a state-of-the-art pooling method. Instead, it is designed to show the simplicity of increasing expressivity by following Theorem 4.4. It shows the general property that contracting a single edge per graph is sufficient to increase expressivity. However, it is only one of many pooling methods satisfying our conditions for increasing expressivity.
> * ClusterPool is an existing method that follows this scheme and has achieved promising empirical results. XP can indeed be seen as a special case of ClusterPool, which highlights the specific property responsible for increasing the expressivity of such pooling methods. We recommend utilizing ClusterPool (or EdgePool), with the novel insight that the reason why these methods increase expressivity compared to node-based pooling methods is shown in Theorem 4.4.

---

> ### Author Response · Authors · 2025-05-06
> **Second answer to reviewer XapJ**
>
> Additionally,
>
> > Why does XP achieve significantly higher performance than ClusterPool on the synthetic task.
> * The threshold above which edges are selected is fixed in ClusterPool. We regularly observed that most edge scores are above or below any fixed threshold, especially when the same threshold is chosen for all layers. Indeed, when selecting only those edges with the maximum score, results become more similar to XP.
> * We added a brief discussion in our experimental section.
>
> > The authors should elucidate how XP can be applied to GNNs in practical settings, considering their non-deterministic and non-differentiable nature.
> * As mentioned above, our goal with XP was to confirm Theorem 4.4. The general property of contracting edges can be incorporated into future pooling methods that may be deterministic and differentiable.
> * As EdgePool and ClusterPool are similarly not fully differentiable in their edge selection, ensuring differentiability of edge-based pooling methods is an exciting future direction, which we view as orthogonal to our insights.
>
> > The authors should validate the performance of various pooling layers on real-world tasks.
> * Apart from XP, all other pooling methods were extensively evaluated on real-world tasks in their respective publications. One of the motivations for our study is that there is no direction (edge-based, node-based, powerful SEL) in graph pooling that significantly achieves better empirical performance. To find promising future directions for research on graph pooling, the ability to increase the expressivity of GNNs is a fundamental property. These directions significantly differ in their ability to increase expressivity. Thus, node-based pooling methods are less promising in that regard.
>
> * Thank you for your suggestion regarding the inclusion of additional real-world experiments. However, we would like to emphasize that our contribution is primarily theoretical. The validity and correctness of our results are grounded in formal proofs and well-defined experimental settings that are sufficient to support our claims. Adding real-world experiments would not strengthen or further validate the theoretical aspects of the paper, nor would it alter the conclusions drawn. Recent results point out that the connection between empirical predictive performance on real world data and expressivity may not be as strong as previously claimed (Morris et al. 2024, Jogl et al 2024) and that increase in performance is likely due to ease of optimization or beneficial inductive biases in the architectures. These views imply that experiments on real world data do not measure what we are interested in: increase in expressivity, while the BREC dataset explicitly measures this.
>
> > Minor Typos:
> * We fixed all the minor typos.
>
> We hope that these revisions adequately address your concerns and improve the clarity and completeness of our work. Should you have any additional comments or suggestions, we would be happy to consider them.
>
> * Christopher Morris, Fabrizio Frasca, Nadav Dym, Haggai Maron, Ismail Ilkan Ceylan, Ron Levie, Derek Lim, Michael M Bronstein, Martin Grohe, Stefanie Jegelka (2024): Position: Future Directions in the Theory of Graph Machine Learning. ICML
> * Fabian Jogl, Pascal Welke, Thomas Gärtner (2024): Is Expressivity Essential for the Predictive Performance of Graph Neural Networks?. SciForDL@NeurIPS

---

### Review · Reviewer_agXC · 2025-04-29

**Summary Of Contributions:**

This article studies the impact of pooling on the expressivity of GNNs. Pooling is defined as a collection of generic operations that scales down the input graph's order by selecting and reducing parts of the graphs, and then connecting the obtained information them in a new smaler graph structure. The pooling operation is summarized in a triplet (SEL, RED, CON).

The authors study when formal expressivity (in the ability to distinguish inputs) is maintained or increased for certain types of pooling.

**Audience:**

Yes

**Claims And Evidence:**

Yes

**Requested Changes:**

Restate the beginning of Part 2 as a formal definitions of GNNs.

I suggest the authors to mention that expressivity here is studied non uniformly (in the case where update functions are neural networks), in the sense that the number of weights of these functions has to grow with the size of the input graphs, to maintain expressivity. For example, it is also the case in (Xu et Al., 2019).

Caption of Figure 3: Change the sentence: ``[...] single superedge between supernodes iff any original nodes were connected results in pooled graphs with different numbers of edges [...]'' (meaning is not clear and syntax incorrect)

**Strengths And Weaknesses:**

Strengths:

1) studying the expressivity of pooling is relevant and useful in practice. Section 4 contains the interesting results, in particular Theorem  4.4 and Proposition 4.6 about examples of pooling operations that can or cannot improve expressivity.

2) The article's contributions are stated very clearly.

3) The article contains a comparison of their results to some existing pooling methods in Section 5.2 and 5.3

Weaknesses: 1) Results of Section 3 are a direct consequence of the existence of a combine-sum function to be injective on multisets (Xu et Al., 2019). Also, the definition of GNNs of the authors allows for any UPD^{t} function (sometimes called combine function). In Section 3, the authors do not discuss if this UPD function is arbitrarily complicated or restricted to MLPs, for example.

2) The insight of Proposition 4.6 is limited in the following sense: It starts with ``Let G1, G2 be two non-isomorphic but WL indistinguishable graphs''. This represents a (very) small proportion of graphs, in particular for applications: the experiments do not showcase limitations or illustrations on datasets for practical applications (only the BREC dataset is used).

3) The computational complexity of pooling is not discussed, in particular the cost of increasing expressivity.

---

> ### Author Response · Authors · 2025-05-06
> **Answer to reviewer agXC**
>
> Dear Reviewer agXC,
>
> Thank you very much for your thoughtful and constructive review of our paper. We appreciate the time and effort you took to provide detailed feedback.
> We have carefully addressed all of your requested changes in the revised version of our manuscript. The modifications are highlighted in the PDF.
> Specifically:
>
> > Restate the beginning of Part 2 as a formal definition of GNNs.
> * We have restated the beginning of Part 2 to include a formal definition of GNNs, as suggested.
>
> > I suggest the authors to mention that expressivity here is studied non uniformly (in the case where update functions are neural networks), in the sense that the number of weights of these functions has to grow with the size of the input graphs, to maintain expressivity. For example, it is also the case in (Xu et Al., 2019).
> * We have now clarified this point explicitly in the last paragraph of Section 3.
>
> > Caption of Figure 3: Change the sentence: ``[...] single superedge between supernodes iff any original nodes were connected results in pooled graphs with different numbers of edges [...]'' (meaning is not clear and syntax incorrect)
> * We have revised the caption of Figure 3 for improved clarity and correctness.
>
> Additionally, we have taken your feedback on the weaknesses into account and clarify on these concerns point-by-point:
>
> > the definition of GNNs of the authors allows for any UPD^{t} function (sometimes called combine function). In Section 3, the authors do not discuss if this UPD function is arbitrarily complicated or restricted to MLPs, for example.
> * In this paper, we consider only neural networks to implement the UPD function and mention this in the preliminaries. The same holds for our M-RED function. In principle, other trainable methods could be employed, as long as they are differentiable, as the overall (pooled) MPNN is trained using SGD.
>
> > The insight of Proposition 4.6 is limited in the following sense: It starts with ``Let G1, G2 be two non-isomorphic but WL indistinguishable graphs''. This represents a (very) small proportion of graphs, in particular for applications: the experiments do not showcase limitations or illustrations on datasets for practical applications (only the BREC dataset is used).
> * Limitations of the 1-WL test and of message-passing GNNs have concrete implications in practical applications. Specifically, many real-world graph datasets require distinguishing graphs based on substructure counts, such as triangles, cycles, or motifs, patterns that 1-WL and so standard MPNNs provably cannot differentiate. For instance, as shown in [Chen et al., 2020], message-passing GNNs are unable to count the number of triangles, which is crucial for graph classification tasks in chemistry and social networks.
>
> > the computational complexity of pooling is not discussed, in particular the cost of increasing expressivity.
> * In this work, we propose expressivity-increasing pooling operators that reduce the size of the pooled graphs. As a first implication, it follows that the computational complexity of MPNN training and inference on this (smaller) graph is bounded by the complexity of the operations on the original graph if the architecture is not changed. The specific methods that we propose as examples are almost all linear time (e.g., identifying connected or biconnected components, filtering node colors). This implies that the computational complexity of building and training the pooled architecture is bounded by the computational complexity of building and training the original architecture. If one employs more expensive select operators (e.g., identifying triangles (Figure 5), which is e.g., O(m sqrt(m)) on sparse graphs), the computational complexity may slightly increase. We have extended our conclusion section accordingly.
>
> We hope that these changes address your concerns. Should you have any further suggestions or comments, we would be happy to incorporate them.
>
> * Chen, Zhengdao, et al. "Can graph neural networks count substructures?." Advances in neural information processing systems 33 (2020): 10383-10395.

---

### Decision · Action_Editor_Xhpx · 2025-06-13

**Recommendation:** Accept with minor revision

**Additional Comments:**

As suggested by one reviewer, Section 5 would be better positioned as a theoretical analysis of existing pooling methods rather than introducing XP as a standalone contribution. Additionally, putting the empirical results in perspective with other baselines would improve the clarity and utility of the experimental section.

**Audience:**

Yes

**Audience Explanation:**

Two reviewers raise important concerns about the lack of practical relevance and the absence of real-world experiments. I agree that the non-deterministic and non-differentiable nature of edge-based pooling methods, as well as the reliance on assumptions shared with Xu et al. (2019), limit the applicability of the proposed approach.

In addition, the performance of XP and the other considered methods on BREC is consistently below all baselines (see Table 2 of the BREC paper), further highlighting the limited practical utility of the approach.

Despite these limitations, the paper meets TMLR’s acceptance criteria: the claims are technically sound and clearly presented, and the topic of understanding the expressivity of pooling methods is definitely relevant to part of the community.

**Claims And Evidence:**

Yes

**Claims Explanation:**

This paper presents a framework that extends the ongoing research thread on GNN expressivity through pooling. The contribution characterizes when pooling layers can increase the expressivity of GNNs. The claims are technically sound and clearly presented and all reviewers agree that the paper makes a legitimate theoretical contribution.

---

> ### Author Response · Authors · 2025-07-14
>
> Dear Editor,
> we have adapted our paper to highlight the theoretical nature of our results as well as the fact that our pooling operator is a proof of concept. We further updated Table 1 to provide a comparison between pooling operators and higher order GNNs, as requested.